# Exploring the Influence of Cognitive and Ecological Dynamics Approaches on Countermovement Jumping Enhancement: A Comparative Training Study

**DOI:** 10.3390/jfmk8030133

**Published:** 2023-09-12

**Authors:** Felice Di Domenico, Tiziana D’Isanto, Giovanni Esposito, Sara Aliberti, Gaetano Raiola

**Affiliations:** 1Department of Human, Philosophical and Education Sciences, University of Salerno, 84084 Fisciano, Italy; fdidomenico@unisa.it (F.D.D.); tdisanto@unisa.it (T.D.); s.aliberti17@studenti.unisa.it (S.A.); 2Faculty of Human, Educational and Sports Sciences, Pegaso University of Naples, 80143 Naples, Italy; gaetano.raiola@unipegaso.it

**Keywords:** countermovement jump, sports training, learning approaches, explosive-elastic strength

## Abstract

Countermovement jumping (CMJ) and free-arm countermovement jumping (CMJFA) express the explosive-elastic force of the lower limbs. Strategies to enhance performance in both types of jumping can be categorized into cognitive and ecological-dynamic approaches. However, the effectiveness of these approaches in improving CMJ and CMJFA remains incompletely understood. This study aims to investigate the impact of training protocols based on the two approaches to improving CMJ. Thirty-six subjects with an average age of 26 years were selected and divided into two groups: the ecological-dynamic group (EDG) and the cognitive group (CG). For 12 weeks, both groups followed separate protocols of three weekly one-hour sessions. EDG group followed a protocol focused on circle time. The CG group followed an instructor-led training protocol. Incoming and outgoing flight heights were measured. Pre and post-intervention differences within and between groups were assessed using t-tests for dependent and independent samples, respectively (*p* ≤ 0.05). CG demonstrated a 12.2% increase in CMJ and a 7.8% improvement in CMJFA, while EDG showed a 10.2% increase in CMJ and 19.5% progress in CMJFA. No statistically significant differences (*p* > 0.05) were observed between the groups in the improvement of CMJ; statistically significant differences (*p* < 0.05) were found in the improvement of CMJFA in favor of EDG.

## 1. Introduction

The assessment of physical efficiency and sports performance provides fundamental data for valid and effective programming of the training process. Kinesiologists, as part of their professional role, utilize various tests aimed at measuring the expressiveness of muscle strength in relation to specific movements. These movements represent constituent elements of complex, specialized skills required for adequate participation in most everyday physical and sporting activities [1]. As such, fundamental movement skills (FMS) typically develop during childhood and refine into specific skills during developmental stages [2,3]. Common FMS include running, throwing, lunging, and squatting, which are transversal to most sports and contribute to maintaining high levels of independence and well-being across age groups [4]. The squat, in both the free-body and overloaded variants, is a multi-joint movement that engages a large number of bones, joints, and muscles of the entire body, particularly the lower limbs [5]. It is used in validated tests designed for assessing strength in this body region. The countermovement jump (CMJ) derives from the squat and entails a vertical jump preceded by a brief leg bend. The ability of the neuromuscular system to rapidly generate maximum concentric strength after an eccentric movement is known as the reactive force regime, or explosive-elastic strength [6,7]. This concept encapsulates muscle performance rooted in the stretch-shortening cycle (SSC), which facilitates the storage and utilization of elastic energy, ultimately leading to increased force during the jumping phase [6,8]. Explosive-elastic strength depends on several factors morphological-physiological factors related to elastic energy storage, preload, muscle activation time, stretch reflexes, and muscle–tendon interactions [9,10,11], as well as coordinative and motivational factors. Elevated levels of reactive force, combined with other factors, enhance efficiency in various scenarios, such as running and directional changes [12].

The countermovement jump (CMJ) can be performed in two distinct variations: bound-arm and free-arm. In the free-arm countermovement jump (CMJ FA), the arms have unrestricted movement, whereas in the CMJ, the arms remain stationary throughout the movement. Both CMJ and CMJ FA are based on powerful movements from which to calculate different variables, such as flight time, height, power, and power relative to body mass, but they have different levels of performance [13,14,15]. Some authors attribute the superior performance observed in CMJ FA to the greater familiarity experienced jumpers have with the movement compared to less practiced individuals [16,17,18]. Conversely, the reduced performance in the CMJ with constrained arms might be attributed to the isolation of lower limb strength production, thereby diminishing potential effects stemming from forces generated in other regions [19,20]. Enhancing reactive strength and associated skills is feasible through training programs focused on various factors, such as the ability to store elastic energy in tendons, the transduction of that energy after a brief isometric period, and the muscle contraction within a specific kinetic chain [21]. Coaches employ two primary learning paradigms to enhance biomotor and sports skills: the cognitive approach and the ecological dynamics approach. The cognitive approach involves instructor/coach-directed exercises aimed at refining motor programs through repetitive task execution. In contrast, the ecological dynamics approach fosters the emergence of functional responses to subjective and contextual demands via self-organization. In this approach, the learner takes center stage, and the coach acts as a facilitator, modifying the environment and executive patterns to encourage creativity, autonomy, and flexibility [22,23,24]. Heuristic-type practicing arising from this process accommodates sudden discontinuous changes in time and space, in addition to linear behaviors [25]. Heuristic practicing is distinct from other methodologies, such as the constraint led approach (CLA) and nonlinear pedagogy, that stimulate learner creativity but do not prioritize the learner’s central role. CLA is rooted in constraints dependent on the interaction of the organism, task, and environment [26]. Nonlinear pedagogy focuses on the environment and task constraints to facilitate exploration, discovery, and functional movement solutions [27,28,29]. The ecological dynamics approach, in contrast, encourages learners to seek autonomous solutions to problems in a variable environment, involving three conditions: altering the learning environment, changing the rules of play, and assuming an observational/evaluative and, if necessary, a group and organizational protective role [30].

The choice of approach must consider basic aspects such as learner needs, training objectives, and environmental characteristics. Skill improvement occurs when learning objectives and teaching strategies are aligned [31]. Depending on their aptitudes, goals, and context, learners improve their skills through two teaching styles: reproductive and productive [32,33]. This evidence suggests a greater adaptability of CMJ-FA to dynamic contexts, as learners are stimulated to respond to the demands of the environment through executive variability by developing a multiplicity of solutions to the desired task outcome [34]. Therefore, environmental and executive variability form the foundation for programming the training process for this gesture. CMJ, being more structured with minimized executive variability, is less transferable to sports performance situations and everyday life but more applicable for maximizing analytical components, specifically the potential reactive strength of lower limbs. Consequently, suitable training processes should focus on automating the gesture. These hypotheses remain unconfirmed in the scientific literature, and the effects of cognitive and ecological dynamics approaches on improving sports-related skills, such as CMJ and CMJ FA, require further investigation. This study aims to examine the effects of two training programs, using cognitive and ecological-dynamic approaches, on enhancing countermovement jumping (CMJ) performance. The investigation includes both constrained and unconstrained arm conditions. The primary objective is to compare the effectiveness of these two approaches and validate or challenge our hypothesis. Specifically, we hypothesize that skills characterized by low executive variability, such as CMJ, would gain a greater advantage from the cognitive approach. In contrast, abilities characterized by greater freedom and variability, such as CMJ-FA, would benefit more from protocols based on the ecological-dynamic approach. In addition, this study explores the impact of these approaches on changes in body mass index.

## 2. Materials and Methods

### 2.1. Study Participants

This study involved 36 participants who were members of a gym located in the Fisciano area (Salerno province). The participants’ ages ranged from 18 to 41 years, with a mean age of 28.15 ± 5.89 years. They all had a minimum of two years of training experience and were in an optimal state of health. Exclusion criteria encompassed potential medical issues or a history of ankle, knee, or back diseases within the three months preceding the commencement of the study. To ensure accurate and representative randomization, a rigorous process was followed. Initially, 18 males and 18 females were randomly selected from the total population of members at a local fitness center. This selection was conducted using a stratified random sampling method, which considered both gender and participants’ physical characteristics. Subsequently, participants were randomly assigned to two research groups: the cognitive group (CG), with an average age of 28.23 ± 6.47 years and an average BMI of 23.08 ± 2.74, and the ecological-dynamic group (EDG), with an average age of 28.06 ± 5.46 years and an average BMI of 23.77 ± 3.21. We ensured that both groups were homogeneous in terms of age and physical characteristics. Furthermore, additional control measures were implemented to ensure that the groups remained comparable throughout the course of the study. These measures included periodic assessment of physical characteristics and analysis of variations in adherence to the fitness program. Participation in the study was voluntary, and informed consent was obtained from all participants.

### 2.2. Procedures

Before initiating the study, participants received comprehensive briefings regarding the tests and procedures involved. Jump heights were assessed using two different methods: the countermovement jump (CMJ) and CMJ-followed by arms (CMJ-FA) tests, both before and after the experimental period. The decision to analyze only CMJ height was based on its established importance in the field of sports science, its solid representation of explosiveness, and the need for a simplified approach to effectively achieve research goals. The initial measurements were conducted seven days after the conclusion of the previous mesocycle and two days before the start of the experimental period. On the same day, the participants’ height and weight were recorded prior to the two tests. The post-experimental period measurements were obtained two days after the experimental period under approximately identical environmental conditions as the initial measurements. Participants adhered to their regular food and fluid intake, refrained from consuming caffeinated beverages for four hours, and abstained from food intake for two hours before the test. Upon completing the initial data acquisition, the two groups underwent training protocols based on distinct approaches for a duration of 12 weeks, involving three weekly sessions. Each training session had an approximate duration of 90 min. Both protocols were structured with a gradual approach to increasing intensity and workload. A systematic progression of weights, repetitions, or sets is incorporated into the plans to facilitate physiological adaptation and enhancement of performance over time. This strategy guarantees that ample stimulus is provided for advancement without compromising the safety and integrity of the training process. In relation to the equilibrium of training volume for the two groups, an adopted approach takes into account the individual maximum strength levels (1 RM) of each participant. The monitoring is achieved using the rate of perceived exertion (RPE). This approach enables the customization of training intensity for each individual, thereby ensuring an equitable distribution of workload between the two groups. The specifics of the two distinct programs are delineated below.

Before undergoing the 12-week experimental period, the entire sample underwent a 2-week anatomical adaptation mesocycle, followed by a 6-week strength development mesocycle. This period ended 1 week before the start of the experimental period. The term ‘anatomical adaptation mesocycle’ refers to a specific phase of training aimed at preparing the body for more intense exercises by focusing on enhancing joint mobility, improving blood circulation, and reinforcing connective tissues [35].

### 2.3. Instruments

A Pegasus professional altimeter scale was used to collect anthropometric data. These anthropometric measurements were recorded on personal identification cards. Jump height assessments were conducted using optoelectronic instrumentation featuring photocells from Microgate’s Optojump system. This instrumentation comprised two transmitter bars along with specialized software for data acquisition and analysis. The execution of the CMJ and CMJ-FA tests was documented using a high-resolution, high-frequency digital video camera (GoPro HERO8). For the implementation of the training protocols, a diverse range of equipment was utilized, including Olympic barbells and discs of varying sizes, kettlebells, medicine balls, and mats.

### 2.4. Data Collection

Entrance tests were conducted after all subjects had had a rest period of one week since their last training and, in any case, two days before the 12-week experimental period. They followed this sequence: measurement of height and weight, measurement of jump height using the Optojump CMJ test, and the CMJ-FA test with image acquisition using a GoPro HERO8 camera. Height and weight measurements were recorded in the Optojump software. The tests were conducted in a prepared room with dimensions 6 × 4 × 4 m (Length × Width × Height) and featured specialized PVC flooring. An acquisition area measuring 90 × 60 cm (Length × Width) was marked on the upper surface of the flooring. The Optojump transmitter and receiver bars were positioned in this area. One of the bars was placed centrally, 50 cm away from the wall at the back of the room. The two bars were spaced 60 cm apart. The GoPro HERO8 camera was positioned on a stable tripod 3 m from the acquisition area. The camera’s focal point was set at a height of 110 cm from the floor, perpendicular to the acquisition plane. Each subject received proper instructions on the acquisition process before the test protocol. Prior to the acquisition protocol, subjects underwent a ten-minute warm-up comprising joint mobility exercises and muscle stretching.

The acquisition protocol consisted of two tests: the CMJ test and the CMJ-FA test. Both tests involved a vertical jump following a phase of lower limb bending and a brief isometric conversion phase. During the CMJ test, subjects kept their hands resting on their iliac crests throughout the trial. Conversely, in the CMJ-FA test, subjects were allowed to move their arms freely in space without any restrictions. Both tests were conducted within the defined acquisition area. Subjects stood upright with their gaze directed towards one short side of the area (aligned with the video acquisition plane). Once positioned in the acquisition area, subjects awaited an acoustic signal from the Optojump program to initiate the jump. Three trials were performed for each test, and the average result was recorded. All data were subsequently logged in the dedicated software.

### 2.5. CG Protocol

The CG protocol involved a mesocycle based on programming managed entirely by the coach using specific drills with the primary goal of improving reactive leg strength and the executive pattern of the gesture. The mesocycle was structured over 12 weeks: initially, more use was made of the contrast method (combination of one strength exercise and one muscle power exercise) and the plyometric method. Each session was characterized by three phases: activation, core, and cool down as shown in Table 1. In the initial phase of 15 min duration, neuromuscular, cardiorespiratory, and joint mobility activation exercises with low to moderate external load were offered. The middle part had a progression of external load application involving contrast methods and plyometrics, as well as a part devoted to the development of other physical parameters. The contrasting method involved the succession of an isometric exercise, wall sits (WS), or a dynamic one, back squat with increasing load up to 80% relative to each subject’s maximal repetition (1 RM) using a barbell, kettlebell, or medicine ball, and an explosive-elastic free-body exercise, CMJ, performed at the distance of 5 s from each other for 3–4 sets, while the recovery between sets was 2 min. The isometric exercise lasted 40–50 s, while the explosive-elastic exercise included 6 repetitions per set. The back squat was used only one day per week, while the CMJ was for the other two. The plyometric method was characterized by the use of drop jump (DJ) from various heights increased every 4 weeks, from 20 to 40 cm, performed for 3–4 sets of 6–8 repetitions. The rest period between repetitions was 5 s, and between sets was 2 min. Finally, the final phase included return-to-normal exercises.

### 2.6. EDG Protocol

A protocol characterized by processes, rather than rigidly defined a priori drills, was administered for the EDG group. A day prior to the commencement of the 12-week experimental period, the EDG participants and their coach convened within the gym center to establish guidelines for the training protocol. This one-hour meeting adopted an approach that positioned each learner at the core of the process, with the coach assuming the role of guide and organizer. The coach defined the meeting’s themes, guided the discourse, and created a secure and stimulating environment to nurture learners’ creativity, autonomy, and adaptability in devising innovative and collaborative solutions.

The meeting was divided into two phases. The initial phase involved a 20 min video presentation showcasing countermovement jumps executed by skilled athletes. Subsequently, the second phase, lasting 40 min, incorporated circle time, a widely employed methodology fostering effective communication and self-expression among participants. In a circular arrangement, both the coach and learners/athletes engage, facilitating mutual attention and inclusivity. As an equal participant, the coach initiated the discussion, encouraged everyone to contribute sequentially, and steered the exploration toward optimal solutions benefiting the entire group. The topic focused on a descriptive analysis of jumping techniques and the pursuit of potential enhancements tailored to individual characteristics. The coach initiated the dialogue, documented all participant-generated solutions, and established overarching guidelines to govern the evolving content of subsequent training sessions.

Each session encompassed several phases as shown in Table 2. An initial 2 min video presentation, selected by the coach, elucidated developmental mechanisms, reactive strength training methodologies, and jump executions, referencing both study participants and accomplished athletes. The subsequent phase involved an 8 min utilization of the educational practice of circle time. During this period, the video was succinctly discussed to delineate the day’s training session based on load progression monitored by the coach. A 10 min activation phase followed, during which athletes engaged in free movements to stimulate neuromuscular and cardiovascular systems. The pivotal third phase consisted of 50 min of training, collaboratively chosen by the coach and learners, adapting load progression and execution dynamics. This was followed by a 10 min cool-down phase. The training load was constantly managed by the trainees through rate of perceived exertion (RPE): for the warm-up and cool-down RPE 3–4, for the central phase RPE 5–7, corresponding to 80% of the 1 RM. Finally, a short circle time segment in which the athletes reported their levels of satisfaction with the session. The coach recorded the responses to inform subsequent sessions.

### 2.7. Statistical Analysis

Descriptive statistics (mean ± SD) were utilized to summarize the data for the different variables. The distribution of each variable was assessed using the Shapiro–Wilk test. To assess differences between input and output data for both groups, the t-test for paired samples was employed. Furthermore, an independent samples t-test was conducted to investigate disparities in the effects of the two protocols. The choice of the t-test was made considering several factors: its simplicity and interpretability, robustness against assumption violations compared to ANOVA, targeted analytical capabilities suitable for our specific comparisons (such as time points within groups), better control over Type I errors associated with multiple comparisons, and alignment with the research question’s focus on meaningful interpretation. All analyses were carried out using SPSS software (version 22; IBM, Armonk, NY, USA), and the significance level was set at *p* ≤ 0.05.

## 3. Results

All subjects completed the planned 12 weeks of training. Table 3 and Table 4 show the results of the CMJ test and CMJ-FA before and after the application of the different protocols in the CG and EDG groups. Improvements in jump height were evident in both tests and both groups: in the CG group, the improvement was 12.21% (2.35 ± 2.44 cm) for CMJ and 7.8% for (1.5 ± 2.24 cm) CMJ-FA; for EDG group there is an improvement in CMJ of 10.22% (1.48 ± 2.89 cm) and in CMJ-FA of 19.54% (3.63 ± 3.79 cm). In addition, there is a variation in BMI in the two groups to consider: for the CG, there is a decrease of 1.5 ± 2.7, and for EDG, there is a decrease of 0.26 ± 1.57.

The descriptive statistics summarize the measurement results using the mean and standard deviation. This allowed us to comprehend that there was an improvement in performance in the two tests across both groups, with varying degrees of enhancement. The protocol grounded in prescriptive teaching exhibited superior results in the CMJ improvement, whereas the protocol aimed at stimulating heuristic learning demonstrated better outcomes in the CMJ-FA improvement. The outcomes of the t-test for dependent samples, as presented in Table 5, conclusively establish the statistical significance of these improvements (*p* < 0.05). Hence, it is evident that these advancements stem from the implementation of both methodologies rather than mere chance. The changes in BMI found in both protocols are not statistically significant (*p* > 0.05).

The differences in results obtained between the two protocols were further processed using a t-test for independent samples to understand whether they had statistical significance in the improvement of CMJ and CMJ-FA. The results shown in Table 6 demonstrate that the difference in improvement obtained between the two protocols in favor of CG (+0.87 cm) in CMJ improvement is not statistically significant (*p* > 0.05), while the better results obtained in the CMJ-FA improvement by the EDG (+2.13 cm) can be attributed to the greater effectiveness of the protocol used by this group (*p* < 0.05). There were no differences between the two protocols in the change in BMI (*p* > 0.05).

## 4. Discussion

The main objective of this study was to investigate the effects of two distinct training protocols, each focused on different learning approaches: cognitive and ecological-dynamic. These protocols were implemented over a twelve-week period to assess their influence on reactive strength in the countermovement jump. The obtained results supported the initial hypotheses, demonstrating that the ecological-dynamic approach was more effective in optimizing performance in the free-arm variation due to its allowance for greater executive variability. The results obtained in the CG can be attributed to the coach’s meticulously designed training program, which involved the thoughtful selection and utilization of specific methods and techniques, along with effective management of the training load. Continuous assessment of individual participant responses (internal load) in each session played a crucial role in achieving this balance [36,37]. Notably, improvements were observed in both the countermovement jump (CMJ) and CMJ with free arms (CMJ-FA), with minimal differences that did not achieve statistical significance (*p* > 0.05). The contrast method and the plyometric method were both found effective in enhancing performance for both types of jumps, aligning with the extensive scientific literature that comprehensively covers these methodologies [38,39,40,41]. The observed adaptations can be attributed to increased neural discharge to agonist muscles and changes in muscle activation strategies, such as improved inter-muscle coordination or alterations in the mechanical characteristics of the muscle–tendon complex. However, these studies primarily focus on describing the neuromuscular or biomechanical effects resulting from specific training protocols, omitting other components that also play a role in the expression of motor and sports skills, notably skill enhancement.

According to the cognitive approach, learning occurs through the repetition of gestures and drills, leading to the consolidation of motor programs within the central nervous system [42,43]. This process unfolds across three stages: cognitive, associative, and autonomous. These stages guide the individual towards enhancing their physical competence, progressively diminishing cognitive processing [44]. The protocol employed in this study placed the coach at the core of the teaching and learning process. The coach prescribed drills aimed at guiding learners to acquire movement patterns (CMJ and CMJ-FA) in alignment with ideal execution techniques. These drills, along with targeted loads, followed a replication learning style, drawing inspiration from the works of Mosston & Ashworth [22] and Goldberger et al. [23]. By adhering to the coach’s instructions over a designated period, participants honed and consolidated their motor skills, gradually reducing cognitive demands. Subsequent post-protocol evaluation revealed heightened familiarity and improved performance in executing both movements.

The results obtained in the ecological-dynamic group (EDG) demonstrate the effectiveness of the proposed protocol in enhancing both CMJ and CMJ-FA (*p* < 0.05). The differences observed between the two movements were minimal and not statistically significant (*p* > 0.05). These findings can be attributed to the inherent characteristics of the protocol, which revolved around the self-organization of the entire group. During the programming phase, the coach actively participated as a member, guiding and overseeing the learning processes. The protocol itself was established based on collective guidelines set before the training period. This approach empowered each group member to contribute to overall performance enhancement, utilizing autonomy, creativity, and flexibility. The integration of video observation and circle time was advantageous, making the study period engaging and motivating for the athletes while also yielding quantitative improvements.

In educational contexts, video observation fosters self-processing, self-determination, and self-regulation within learners, all without direct coach intervention [24]. This approach allowed individuals to observe correct execution while identifying potential execution errors and independently devising strategies to rectify these errors [45]. Our study presented the CMJ and CMJ-FA videos performed by accomplished athletes from various disciplines, along with recordings of entrance tests conducted on each EDG subject. The aim was to encourage students to identify errors and strengths in comparison to internal elements (EDG components) and external subjects (skilled athletes) when performing actions, generating potential solutions for reinforcement. The central focus of the entire experimental period of EDG was the utilization of circle time. During this phase, learners and coaches engaged in peer discussions based on insights gleaned from previously viewed videos. Together, they autonomously set goals and collaboratively brainstormed solutions through peer interaction [46]. Previous research has shown the effectiveness of this approach in enhancing social skills and academic performance [47,48,49], and our study’s positive outcomes align with these findings. Regarding the final finding of the study, the difference in effectiveness between the two protocols concerning the CMJ and CMJ-FA improvement, we found a statistically significant difference (*p* < 0.05) in favor of the EDG group only for CMJ-FA. For CMJ, there was no statistically significant difference (*p* > 0.05). These results confirm our research hypotheses, demonstrating that the ecological-dynamic approach favors movements characterized by greater executive variability [50], while the cognitive approach is more suitable for controlled abilities with few environmental variables. In the CMJ-FA, participants were given minimal directions on arm usage, allowing them to interpret and search for the best solution for optimal performance. In the CMJ, however, the entire gesture allowed no variation from the initial directions. Thus, the protocol proposed according to the cognitive approach probably produced analytical adaptations related to improving the expression of lower limb strength with minimal interference from other external forces. These training protocols could be more effective if they were associated, especially in the phase of the approach to performance, with specific work in which the skills are solicited in different contexts similar to the situations verifiable in the specific competition. Whereas, in the case of the protocol based on the ecological-dynamic approach, the improvement that resulted was attributed to several factors: in the first instance, there was a structural and coordinative adaptation to be attributed to the load applied and protracted over time (12 weeks). However, the most decisive factor was the continuous stimuli that induced the participants to self-organize themselves according to self-determined schemes to respond with innovative and functional solutions to the context’s requirements. This hypothesis is corroborated by the result obtained in the improvement of CMJ-FA, which was greater in EDG, whereby by increasing executive variability, the subjects were more stimulated to come up with innovative and increasingly effective solutions to achieve the performance objectives.

In this study, we also included BMI values in order to see if, in addition to the change in performance levels, there was also a change in parameters related to the subjects’ well-being and health. However, the analysis of the BMI values did not uncover any significant changes either within the groups or in the comparison between them. This observation could potentially be attributed to the characteristics of the entire sample. Notably, the baseline BMI values fell within the normal range, specifically within the range of 20 to 24.99 [51]. Consequently, it is plausible that the impact of the two protocols on the groups was negligible due to these baseline conditions. Additionally, the consistent RPE values across both groups suggest a uniform perception of exertion during the tasks. However, it is crucial to acknowledge the possible influence of external factors and individual variabilities that might have contributed to this consistent perception. Variables such as the environmental conditions during task execution and the inherent traits of the participants might have contributed to maintaining this equilibrium. It would be interesting to expand the study to subjects with different BMIs.

This study has several limitations, including the lack of other studies dealing with the application of methodologies based on the two learning approaches to transferable FMS to sports skills and the relatively small sample size. The absence of a control group is another important limitation to acknowledge in this study. Therefore, further investigations that include a control group are needed to clarify these theories on larger samples, allowing us to draw more robust conclusions about the relationships and effects observed in this study.

## 5. Conclusions

After 12 weeks of training, both protocols improved performance in CMJ and CMJ-FA, although in different ways. The usefulness of this study lies in having helped shed light on how the two different approaches are applied. Methodologies based on the cognitive approach follow structured programs that aim to manage external load versus internal learner responses. Methodologies based on the ecological dynamics approach, on the other hand, follow processes that stimulate the emergence of functional responses useful for achieving certain learning goals. For this reason, kinesiologists, personal trainers, and coaches, in planning the training process for performance improvement, must consider multiple aspects, including the ultimate goals, the starting situation, and the characteristics of their athletes so that they can choose the appropriate teaching/learning methodologies.

## Figures and Tables

**Table 1 jfmk-08-00133-t001:** Detailed description of the 12-week CG protocol, including phases, description, exercises, number of sets, and repetitions.

Phase	Description				Load	Duration
Initial phase	Neuromuscular and cardiovascular activation with joint mobility exercises				Low to moderate	15 min
Central phase	Introduction of methodologies, tools, and content aimed at enhancing reactive strength	Weeks 1–12	Exercises		Moderate to high (40–85% di 1 RM)	50 min
		1–4	Contrast method. Wall sit * + CMJ	Sets: 4 × 40 s + 6 repetitions Rest: 2′		
			Agility: Hops in-out in forward and change of direction with ladder	Sets: 3 × 20′’ Rest: 1′		
			Strength and hypertrophy **:	Sets: 4 × 12 − 10 − 8 − 6		
		5–8	wall sit with 5 kg medicine ball + CMJ	Sets: 3 × 40 s + 6 repetitions		
			Drop jump ***	3 × 6 repetitions		
			Agility: X drill	Sets: 3 × 20′’		
			Strength, hypertrophy ****	Sets: 3 × 8		
		9–12	wall sit with 5 kg medicine ball + CMJ	Sets: 3 × 40 s + 8 repetitions		
			Drop jump ***	6 × 6 repetitions		
			Ladder skip in forward	Sets: 3 × 20′’		
			Strength, hypertrophy *****	3 × 10		
Cool down	De-fatigue and stretching exercises				Low load	10 min
* Week 1 performed free-body, from week 2 performed with 5 kg medicine ball. ** Fundamental exercises performed with increased intensity over individual 1 RM: day 1 back squat, day 2 flat bench, day 3 barbell rower. *** The drop height corresponds to the maximum jump height achieved in the CMJ test. **** Exercises performed at 80% of 1 RM: day 1 barbell deadlifts; day 2 flat bench press dumbbell stretches; day 3 pull-ups. ***** Exercises performed at 75% of 1 RM: Day 1 Bulgarian squats; day 2 oblique bench presses with dumbbells; day 3 chin-ups.						

**Table 2 jfmk-08-00133-t002:** Detailed description of the 12-week EDG protocol, including steps, methods, description, and duration of processes.

Phases	Method	Description	Duration
Initial phase	Video viewing	Watching short videos showcasing various jumping techniques performed by skilled athletes or group members during the entry test.	10 min
	Circle time	Athletes and the coach gather to identify issues and goals. Solutions are devised to build upon previous work or propose new strategies while adhering to the plan.	
Warm-up	Joint mobility, moderate-intensity work	Engaging in joint mobility exercises and moderate-intensity activities to activate cardiovascular and neuromuscular systems and promote joint lubrication.	10 min
Central phases	Strength expression exercises (varied by period)	Employing specific methodologies and tools with varying intensities to enhance different aspects of strength and core stability. Decisions are made through circle time, based on the initial program.	50 min
Cool down	De-fatigue and stretching exercises	Performing exercises to alleviate fatigue and engage in stretching routines, aiming to restore the body’s equilibrium.	10 min
Final phase	Circle time	A brief meeting to assess the session’s strengths and weaknesses.	5 min
The equation to calculate the training volume is Volume = Number of sets × Number of repetitions × Weight lifted (80% of 1 RM).			

**Table 3 jfmk-08-00133-t003:** Pre and post-protocol test results in the CG group.

						CMJ Pre	CMJ Post	CMJ-FA Pre	CMJ-FA Post	Pre-Post Difference in Jump Height			
	Age	Weight (kg)	Height (m)	BMI Pre	BMI Post	Jump Height (cm)	Jump Height (cm)	Jump Height (cm)	Jump Height (cm)	CMJ		CMJ-FA	
										cm	%	cm	%
Mean	28.23	67.12	1.7	23.07	22.61	23.28	25.63	26.54	28.04	2.35	12.21	1.51	7.78
SD	6.47	10.41	0.05	2.73	1.85	6.81	6.57	7.42	6.34				

**Table 4 jfmk-08-00133-t004:** Pre and post-protocol test results in the EDG group.

						CMJ Pre	CMJ Post	CMJ-FA Pre	CMJ-FA Post	Pre-Post Difference in Jump Height			
	Age	Weight (kg)	Height (m)	BMI Pre	BMI Post	Jump Height (cm)	Jump Height (cm)	Jump Height (cm)	Jump Height (cm)	CMJ		CMJ-FA	
										cm	%	cm	%
Mean	28.06	70.18	1.72	23.77	23.5	23.28	24.76	27.19	30.83	1.47	10.22	3.64	19.64
SD	5.46	10.31	0.07	3.21	1.93	8.12	6.67	9.89	11.05				

**Table 5 jfmk-08-00133-t005:** Student’s t-test results for dependent samples show differences after 12 weeks of applying both protocols for both tests.

Group	Test	Average		t-Value	*p*
		Pre	Post		
CG	BMI	23.07	22.61	1.28	0.22
	CMJ	23.28	25.63	−4.08	0.01
	CMJ-FA	26.54	28.04	−2.85	0.05
EDG	BMI	23.77	23.5	0.72	0.48
	CMJ	23.28	24.76	−2.17	0.05
	CMJ-FA	27.19	30.83	−4.08	0.01

**Table 6 jfmk-08-00133-t006:** Student’s t-test results for independent samples to show differences in outcomes after 12 weeks between the two groups.

Test	Group	Average	Variance	t-Value	*p*
BMI	CG	−1.15	7.32	−1.21	0.24
	EDG	−0.27	2.45		
CMJ	CG	2.35	5.97	0.98	0.33
	EDG	1.48	8.36		
CMJ-FA	CG	1.51	5.01	−2.06	0.05
	EDG	3.64	14.35		

## Data Availability

Not applicable.

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
