# Peer review of "Exploring the Influence of Cognitive and Ecological Dynamics Approaches on Countermovement Jumping Enhancement: A Comparative Training Study"

_jfmk, 2023, doi:10.3390/jfmk8030133_

Round 1
Reviewer 1 Report
Coment 1
In the introduction there is a lot of talk about learning and practicing. These are two different things. In my opinion, the emphasis in the introduction should be on practicing. Jumps are biotic motor skills and are needed in daily life from early childhood. The respondents have been practicing for at least two years, and I assume they have learned various jumps during that time. This is about performance improvement or practice.
Line 107-109
Thus, this study aims to verify the effects of two training programs based on the two learning approaches on improving countermovement jumping with both constrained and free arms and to compare the effectiveness of the two approaches.
Coment 2
Can we even speak of learning here? The oldest respondent is 41 years old and should have learned to jump in childhood. We can talk about practise or correction of motor errors. Learning refers to the acquisition of new motor skills.
Line 118
The sample was randomly divided into two groups, each comprising 18 individuals. These groups consisted of 9 males and 9 females each.
Coment 3
Please clarify the randomization with regard to the same number of men and women, same age and physical characteristics.
Line 136-139
Upon completing the initial data acquisition, the two groups underwent training protocols based on distinct approaches for a duration of 12 weeks, involving three weekly sessions. These protocols were introduced after a 2-week anatomical adaptation mesocycle, followed by a 6-week strength development mesocycle. Each training session had an approximate duration of 90 minutes.
Coment 4
2-week anatomical adaptation mesocycle + 6-week strength development mesocycle = 8 weeks
4 weeks are mising
Line 292-294
However, these studies primarily focus on describing the neuromuscular or biomechanical effects resulting from specific training protocols, omitting other components that also play a role in the expression of motor and sports skills, notably the learning aspect.
Coment 5
How to examine the learning aspect of a very simple motor task (jump) that the subjects already knew. I would like to mention that two days before the protocol they conducted an initial test, how did they test something they had not yet learned?
Final coment
This study is technically correct, but it examines practice rather than learning. I think the respondents had to learn a new content or sport. The result of learning is new motor skills, and the result of practice is improved performance.
Author Response
Dear Reviewer,
thank you for reviewing our manuscript. We appreciate it. We have followed your suggestions point by point to improve the manuscript quality, according to our possibilities. Changes have been made to the full text using word tracking to detect changes immediately. Thanks for your time.
AUTHORS (A)
REVIEWER 1 (R1)
R1: Comment 1 - In the introduction there is a lot of talk about learning and practicing. These are two different things. In my opinion, the emphasis in the introduction should be on practicing. Jumps are biotic motor skills and are needed in daily life from early childhood. The respondents have been practicing for at least two years, and I assume they have learned various jumps during that time. This is about performance improvement or practice.
A: Dear Reviewer, thank you for reviewing our manuscript. We appreciate your suggestion. We have rewritten the introduction following your suggestions, placing more emphasis on practice rather than learning.
R1: Comment 2, Line 107-109 - Can we even speak of learning here? The oldest respondent is 41 years old and should have learned to jump in childhood. We can talk about practise or correction of motor errors. Learning refers to the acquisition of new motor skills.
A: We accepted the suggestion by providing a rewording of the sentence
R1: Comment 3, Line 118 - Please clarify the randomization with regard to the same number of men and women, same age and physical characteristics.
A: We proceeded to clarify the randomization.
R1: Comment 4, Line 136-139 - 2-week anatomical adaptation mesocycle + 6-week strength development mesocycle = 8 weeks
A: Thank you for the observation. I was not clear in the wording of the sentence. The 12-week experimental period was preceded by an 8-week period of anatomical adaptation (2 weeks) and strength development (6 weeks). We have modified the sentence.
R1: Comment 5, Line 292-294 - How to examine the learning aspect of a very simple motor task (jump) that the subjects already knew. I would like to mention that two days before the protocol they conducted an initial test, how did they test something they had not yet learned?
A: Accepting the suggestion, we rephrased the sentence by replacing the term "learning" with "skill enhancement". By changing "learning" to "skill enhancement," the sentence clarifies that the focus is on measuring improvements or refinements in the subjects' existing skills, rather than learning something entirely new. This adjustment maintains the coherence of the context and the study's methodology.

Reviewer 2 Report
Exploring the Influence of Cognitive and Ecological Dynamics approach Approaches on Countermovement Jumping Enhancement: A Comparative Training Study
This is an interesting study where Authors apply a cognitive preparatory period before training in order to enhance CMJ and CMJFA performance. Although Authors have done a great training study (12 weeks) with a fair number of participants (N=36) there are some major concerns that need to be addressed and some structural revisions that need to be answered.
Main concerns:
1. Why T-Test statistical analysis was chosen? From the experimental design it looks like a 2x2 (time x groups) repeated measures ANOVA.
2. Why participants were randomly assigned into two groups? Normally, participants should be separated according to CMJ performance in order to have an initial balance performance.
3. A control group is missing here. This should be pointed out into the limitations.
4. How Authors control-balanced the training volumes of the two groups? Also, details are needed regarding the progressive overload of the programs.
5. Twelve weeks of training and two different training programs only to measure CMJ height? And how Authors will explain the findings or the training-induced changes from the programs on CMJ?
Abstract:
Abstract is not following the instruction for Authors of the journal and needs to be re-written (word count=200 words). Please, keep the statistical indexes inside the abstract.
Introduction:
Introduction needs more references. There many statements where references are crucial in order to support the sentences.
Lines 45-47: Add a reference.
Lines 47-49: Add a reference, not so sure that reference 2 is appropriate here.
Lines 49-51: I am not convinced that during the squat exercise the lower muscles are the only muscles involved.
Lines 53-55: Add a reference.
Lines 59: Reference is too old. Please, find a modern reference.
Line 65: Both CMJ and CMJFA are powerful movements from which a sport scientists or a kinesiologists may calculate several variables like the flight time, height, power, power relative to body mass, etc. I am not convinced that CMJ may be expressed as reactive strength. In contrast, reactive strength can be evaluated from isometric mid-thigh pulls force-time curve, drop jumps and other faster movements compared to CMJ.
It is difficult to lead the readers to the research question since there is a great absent of relative references.
Lines 107-109: Clearly state the purpose of the study. Add a hypothesis.
Methods:
Line 115: Add a SD.
Line 120: This is a cognitive group, not a control group. Please correct this throughout the manuscript.
Line 120-121: Add indexes of measurement for BMI.
Lines 138-140: Please explain the term “anatomical adaptation mesocycle”. If it’s needed add a reference.
If I am correct, all participants followed 2 weeks of anatomical training and another 6 weeks of strength training before entering the 12 weeks experimental period. If this is the case, an experimental figure would be helpful.
Data collection: Why only the CMJ height was used in the study? We have seen many times that although non-significant changes occurred in height strong changes can be found in power and power relative to body mass. This really limits the whole study. Moreover, 12 weeks of training only to measure CMJ height weakens the power of the study. More variables like 1RM strength, power, sprint, may significantly contribute to nature of the results.
Line 187: What does the 80% represents? How the 80% was calculated and for which exercise? Authors should really work on the program presentation. Table 1 is problematic; Authors can add the actual exercises names and not abbreviations Ex.1+Ex.2. Also, details are missing like the exact agility exercises and the explanation of the “explosive elastic exercise”.
Table 1: How Authors define the contrast training in their study?
In Table 2 the training volume of the central phase must be balance with the CG. Authors should present the equation of training volumes.
Results:
Results are poor. Only cm and % for CMJ and CMJ FA while in the same time SDs are missing inside the text.
Discussion:
Only with CMJ height the discussion around the reactive strength is limited.
Line 277: Delete “squat”.
Line 280: Either cognitive group or control.
Lines 283-284: No results from internal load are presented here. Then how Authors explain changes in performance?
Lines 285-286: No need to provide the abbreviations again. Instead, apply the abbreviation throughout the manuscript.
Authors present the results of the study but there is a need to dig deeper in order to explain the findings. Readers, should take the message easily for the study.
Line 351: I am not sure I followed the sceptical of the Authors regarding the different ways of CMJ enhancement.
Several questions are rising: Was there an increase in body mass, in lean mass, in 1-RM strength or in power and agility? Τhe study will be greatly strengthened by the addition of such information
Round 2
Reviewer 1 Report
The essence of the article has been changed, it can be published in this form.
Author Response
Dear Reviewer,
Thank you for your feedback and for taking the time to review our article. We appreciate your insights and comments.
We are pleased to hear that you find the revised version of the article suitable for publication. Your feedback has been valuable in shaping the content and direction of the manuscript. We have carefully considered your suggestions and made the necessary changes to align the article with the intended scope and objectives.
Sincerely,
Reviewer 2 Report
1. Add the reason of T-Test analysis in the text.
2. Add the reason why 1 only parameter was analyzed from the CMJ in the methods.
3. Strengthen the discussion with the possible mechanisms which enhanced CMJ height.
